# Polarization-Doped InGaN LEDs and Laser Diodes for Broad Temperature Range Operation

**DOI:** 10.3390/ma17184502

**Published:** 2024-09-13

**Authors:** Muhammed Aktas, Szymon Grzanka, Łucja Marona, Jakub Goss, Grzegorz Staszczak, Anna Kafar, Piotr Perlin

**Affiliations:** 1Institute of High Pressure Physics “Unipress”, Sokolowska 29, 01-142 Warsaw, Poland; maktas@unipress.waw.pl (M.A.);; 2Proinspiria Jakub Goss, Rubinowa 41, 05-500 Piaseczno, Poland

**Keywords:** optoelectronics, laser diode, LED, cryogenic temperature, III-nitride semiconductor, InGaN quantum wells, polarization doping

## Abstract

This work reports on the possibility of sustaining a stable operation of polarization-doped InGaN light emitters over a particularly broad temperature range. We obtained efficient emission from InGaN light-emitting diodes between 20 K and 295 K and from laser diodes between 77 K and 295 K under continuous wave operation. The main part of the p-type layers was fabricated from composition-graded AlGaN. To optimize injection efficiency and improve contact resistance, we introduced thin Mg-doped layers of GaN (subcontact) and AlGaN (electron blocking layer in the case of laser diodes). In the case of LEDs, the optical emission efficiency at low temperatures seems to be limited by electron overshooting through the quantum wells. For laser diodes, a limiting factor is the freeze-out of the magnesium-doped electron blocking layer for temperatures below 160 K. The GaN:Mg subcontact layer works satisfyingly even at the lowest operating temperature (20 K).

## 1. Introduction

Nitride-based semiconductors are one of the primary semiconductor materials used for the fabrication of light emitters due to the wide range of available energy bandgap (InN Eg = 0.7 eV, GaN Eg = 3.4 eV, AlN Eg = 6.2 eV) and direct bandgap of all the binary compounds. Since the early 1990s, Mg doping has been successfully used in optoelectronic bipolar devices for creating p-type material, being a critical factor for the proliferation of solid-state lighting technology. The current popular nitride semiconductor applications include white LEDs [1,2], displays [1,3], projectors [4,5], car headlights [6,7], and Blu-ray standard [8] (although the peak of this technology has certainly passed). The development of nitrides also opened a path for novel advanced technologies like underwater communication [9], wireless internet connection by visible light (Li-Fi) [10], and atomic clocks [11,12]. 

However, even with the successful commercialization of various nitride devices, there is one material aspect of the state-of-the-art technology that is far from optimal. Due to the low dielectric constant and high effective mass of holes in AlGaN/GaN semiconductors, the Mg acceptor has a high thermal activation energy of around ~170 meV for GaN and ~600 meV for AlN. This high value means that, at room temperature (~26 meV), only a relatively small number of Mg atoms are ionized. Moreover, the hole concentration in Mg-doped GaN changes dramatically with temperature. There are many reports on the dependence of activation energy and hole concentration on both doping concentrations and temperature [13,14,15,16,17]. Arakawa et al. showed how hole concentration changes with temperature for different doping levels of samples [15]. The hole concentration decreased by approximately a factor of 10^3^ from room temperature to 165 K. This dramatic change in hole concentration directly affects the electrical conductivity and injection efficiency of the p-type material.

Although nitride devices have serious drawbacks in the aspects of operation in low temperatures, there are still reasons to pursue such a goal. First, we consider direct applications in rough environment conditions. One of the examples is visible light communication in space applications that in some cases requires operation at very low temperatures. Second, the laser diode can be placed inside of the cryostat to work as an excitation light source in scientific studies. Lastly, the polarization-doped structure might be the only option for studying temperature-dependent material parameters like as material gain at cryogenic temperatures. Overall, an LED and LD that can work at cryogenic temperatures can help to reach new technological applications and material science.

Many research groups are interested in extending the usable temperature range of nitride semiconductor devices, particularly towards a low temperature range. Hori et al. presented single quantum well (QW) green and blue LED electroluminescence (EL) measurements down to 20 K [18]. The integrated EL intensity of both samples decreased after 180 K, which is around the hole freeze-out temperature for GaN. Additionally, Cao et al. presented similar investigations for green, blue, and UV LEDs, observing the same behavior for blue and UV LEDs (decreasing EL intensity after around 150 K), but the EL intensity of green LEDs did not decrease with varying temperatures [19]. They indicated that the green LED’s higher In concentration increases localization effects in carrier capturing and recombining processes.

Grzanka et al. showed how a highly doped electron blocking layer (EBL) affects EL characterization at cryogenic temperatures [20]. The integrated EL intensity decreases below 150 K for samples with the EBL layer, while the integrated EL intensity almost does not change for samples without the EBL layer. Lee et al. demonstrated how external quantum efficiency (EQE) changes with decreasing temperature for various injection currents [21]. The EQEs steadily increased up to 200 K and then decreased rapidly. Moe et al. performed low-temperature measurements of ultraviolet (UV) LEDs, demonstrating 5 mA continuous wave (CW) operation between 8 K and 300 K, with the intensity increasing until approximately 120 K and then decreasing suddenly [22]. In all these studies, except for Cao’s green LED, a rapid drop in intensity has been observed due to the Mg freeze-out zone, which occurs at around 150 K.

While Mg remains most likely the only chemical acceptor effective in GaN and AlGaN, there is an alternative solution originating from the crystalline structure of III-N materials. AlInGaN semiconductors crystallize in the hexagonal wurtzite structure—the lattice that lacks an inversion symmetry. This allows for the presence of dielectric (spontaneous) polarization in the material even without any strain (pyroelectric effect). An additional factor, which makes both spontaneous and piezoelectric polarization particularly large, is the highest ionicity among the group III–V semiconductors [23]. The existing polarization creates a fixed charge density at the interfaces of all the heteroepitaxial layers and is the cause of a notorious quantum-confined Stark effect (QCSE) [23,24], which consists of the spatial separation of electron and hole wave functions. The polarization charge attracts free carriers of the opposite charge, creating a two-dimensional electron gas (2DEG) or two-dimensional hole gas (2DHG). These two-dimensional carrier gases have high mobility and high carrier concentration and are the core idea of high electron mobility transistors (HEMTs). Furthermore, if we fabricate a nitride semiconductor with a composition gradient, we should expect the appearance of three-dimensional fixed charges and, consequently, three-dimensional carrier gases: three-dimensional electron gas (3DEG) or three-dimensional hole gas (3DHG). This approach is known as polarization doping.

Polarization doping is explained in detail in our previous study [25] and many other papers [26,27,28,29,30,31]. Densities of the three-dimensional carrier gases can be calculated using Poisson’s equation: ρ(z)=−∇·P(z), where z is the growth direction (in our case, the c-direction), ρ(z) is the carrier concentration of the three-dimensional gases, P(z) is the z component of the polarization density vector, and ∇· is the divergence operator. To replace Mg-doping with 3DHG, the Al composition should be decreased along the c-direction. P-type polarization doping can also be achieved by grading InGaN, but it is much more practical to use AlGaN graded layers in applications such as laser diodes.

One advantage of polarization doping is that the carrier concentration should not be affected by temperature. This expectation is related to the fact that the fixed charge density is obviously temperature independent. It is less clear whether the attracted mobile charge density is temperature independent, but it appears to be the case as reported by several groups. Jena et al. demonstrated the carrier concentrations of 2DEG and 3DEG between 20 K and 300 K [32]. The concentration was almost constant across this temperature range. Simon et al. presented a temperature-dependent 3DEG carrier concentration between 10 K and 300 K, created by fabricating AlGaN layers of identical maximum Al content but with different thicknesses [33]. The carrier concentration decreased with increasing thickness but remained almost constant with temperature changes. Additionally, Simon et al. showed that the carrier concentration of Mg co-doped 3DHG changed with temperature between 4 K and 300 K [26]. The structure was grown on an N-polar face substrate. A slight decrease in concentration was observed, but no freeze-out of holes occurred. Zhang et al. also demonstrated the co-doped layer of Ga-polarity (Mg plus polarization doping) [34]. Temperature-dependent changes in carrier concentration were observed between 100 K and 300 K. The difference between this work and other studies is that there was a decrease in carrier concentration, but it was limited to low temperatures. 

In this study, we performed low-temperature measurements of LEDs and LDs with polarization-doped p-type layers. The devices (both LEDs and LDs) were characterized by measuring electroluminescence (EL) spectra and light-current-voltage (L-I-V) characteristics as a function of temperature. Additionally, for the LDs, we measured the temperature-dependent threshold current and slope efficiency. 

## 2. Materials and Methods

The polarization-doped p-layer LED and LD structures are shown in Figure 1. Both structures are based on a laser design, consisting of an AlGaN cladding layer and an InGaN waveguide layer. The modification occurs in the layers above the quantum wells. The LED structure has a top layer of AlGaN that is thinner than in the case of the LD. Please note that the LED structure contains magnesium only in the subcontact layer to lower the Schottky barrier at the interface between the metal and semiconductor. The structure and the optimization of polarization-doped laser diode are described in detail in our earlier publication [25]. 

In Figure 2, we demonstrated an energy band diagram and a refractive index of the laser structure (calculated by SiLENSe v6.4). The built-in electric field strongly modifies the energy band diagram, which can be observed, e.g., in the shape of the EBL region. Please note the potential spikes at the n-side edge of the gradient and at the n-side edge of the EBL. These appear due to the discontinuity of the polarization at the interfaces between layers and may prevent the transfer of holes to the active region. In Figure 2b,c, we present the magnified graphs in terms of x axis in the areas of active region and EBL region. The design of the graded layers was done using a simple formula expressing the relationship between the density of polarization charge, and the composition gradient is given as follows: $\rho \left(z\right)\approx 5\times 10^{13}\times \left(x_{1}-x_{2}\right)/d\;\left[{\mathrm{cm}}^{-3}\right]$, where $x_{1}$ is the initial Al concentration of the gradient layer, $x_{2}$ is the final Al concentration of the gradient layer, and d is the thickness of the layer given in cm [32]. In our samples, we use polarization-doped layers with a carrier density at the level of 1.1 × 10^18^ cm^−3^ for an LED structure and 1.9 × 10^17^ cm^−3^ for an LD structure [20].

As we explained in our previous work [25], the electron blocking layer (EBL) is Mg-doped to improve the injection efficiency of this device. In the LED structure, a staggered quantum well is used. This method is common in high-In-content structures to improve the overlap of an electron and hole wavefunctions [35,36]. However, the benefit is negligible for low-In-content QWs (blue and shorter wavelengths). In an LD structure, we decided to use the simpler, classical rectangular quantum well. The difference in the quantum well design between the LED and the LD is not related to the studies within this paper and should not influence the conclusions in any way. 

The samples are grown by Metalorganic Vapor Phase Epitaxy (MOVPE) on an n-GaN bulk substrate, and the material parameters are checked by x-ray diffraction. LDs are produced as oxide-isolated ridge waveguide devices. The mesa formation is made by using reactive ion etching (RIE). The mesa is etched down to approximately 50 nm before reaching the EBL layer. The ridge width is around 1.8 µm, and the resonator length is around 750 µm. For the insulation layer, SiO_2_ is deposited by Inductively Coupled Plasma Chemical Vapor Deposition (ICP CVD). Ti/Al/Ni/Au ohmic contact is used for the n-side to the back side of the highly conductive n-GaN substrate. For p-side ohmic contact, Ni/Au is coated on top of the ridge structure. The LD is mounted on a TO-56 package, and the LED chip is mounted on a copper plate. 

## 3. Results

First, we would like to demonstrate the performance of the polarization-doped LED. We performed continuous operation temperature-dependent spectral and I–V measurements between 300 K and 20 K, which are given in Figure 3. The LED successfully operated even at 20 K. The operating voltage increased with decreasing temperature, e.g., for 100 mA from 3.9 V (300 K) to 5.2 V (20 K), as shown in Figure 3a. This change is probably related to the partial freezing-out of acceptors in the subcontact layer. However, due to the self-heating in this area of the sample, there is still a considerable amount of acceptors ionized, allowing for the electrical injection and operation of the device.

In the EL measurement, shown in Figure 3b, we observed that the intensity increased gradually. The peak wavelength showed a blue shift with decreasing temperature, which is expected as a consequence of bandgap increase but moderated by the carrier localization. It is worth stressing that even at the lowest ambient temperature (20 K), the polarization-doped LED could still operate at a high current of 100 mA. The integrated EL intensity measured as a function of temperature is shown in the inset of Figure 3b. Please note that the integrated EL was estimated by integrating only the part of the spectra that correspond to the QW emission. The emission intensity increases down to 100 K and then stays almost constant down to the lowest temperature. This behavior corresponds to a decrease in Schottky–Hall–Read (SHR) nonradiative recombination at low temperatures. Below 200 K, we started to observe a secondary emission peak with energy corresponding to the top InGaN layer. A similar phenomenon was observed by Chlipala et al. [37] and explained by the decrease in the EBL barrier due to the free-hole freeze-out and the consequent electron leakage into the InGaN layer above the quantum well. Although the overall mechanism here is similar, we do not have magnesium-doped EBL, so we would rather associate the observed emission with an enhancement of the electron overshoot as a consequence the worsening of hole transport at low temperatures (e.g., mobility edge).

In the case of the laser diode, we conducted temperature-dependent EL and light-current (L-I) measurements from 295 K down to 77 K in both pulsed current and continuous wave (CW) modes. The reason for the difference in temperature range between LED and LD structures is technical limitations, as we performed the measurements in two different setups. During both experiments, the temperature is measured by a thermocouple connected to the cryostat’s cold finger, which should be the same as the laser package. In the case of CW measurement, the indicated temperature may be significantly different from the temperature of the lasers’ junction and the subcontact area due to Joule heating. This issue is discussed further in the paper. Due to the lack of possibility of precisely estimating the junction temperature, all the graphs present the ambient temperature. 

Pulsed current was employed to mitigate device heating from Joule heating (self-heating). This is crucial because, at cryogenic temperatures, the cryostat’s cooling capacity can approach the energy dissipation within the device. For our experiments, we utilized the ILX Lightwave LDP3811 Precision Pulsed Current Source, which provided 200 ns pulses at a 1 kHz repetition rate, resulting in a duty cycle of 0.02%. Our experience suggests that, for such short pulses and small duty cycle, the self-heating effect becomes negligible. The temperature-dependent measurement results are presented in Figure 4.

Down to 200 K, the laser parameters (threshold current and slope efficiency) remain satisfactorily stable. The small tendency observed in Figure 4b can be explained by two mechanisms. The first is the lowered effectiveness of the Mg-doped EBL [38] by lowering the temperature. The reason is the strong dependency of the Mg ionization on temperature, as discussed in the introduction. We further discuss the issue in more detail in the text. The second is that the low mobility of holes in nitride materials makes it difficult for them to reach the active region (they may be stuck on potential barriers). Lowering the temperature further reduces the efficiency in passing those barriers. For calculating slope efficiency, almost the whole curve above the threshold is used for linear fitting. Non-linearities are present above the threshold current, but we chose the linear fitting method to obtain an average slope value for each measurement. Below 180 K, the threshold current increases rapidly, while the slope efficiency decreases, ultimately leading to a cutoff of laser operation below 160 K. Because of the cutoff, we cannot estimate any threshold current and slope efficiency below 160 K.

The curves have an exponential shape, which suggests that the device is working in the amplified spontaneous emission regime (amplification is present), but still, there are not enough carriers to reach lasing.

The spectral characteristics of the studied laser diode (LD) are shown in Figure 5 below, specifically in Figure 5a,c. These figures illustrate the evolution of spontaneous emission with temperature. It is clearly visible that, with decreasing temperature, the emission shifts towards higher energy, though much less than expected for the InGaN bandgap. This is a typical demonstration of “S-shape” behavior commonly associated with carrier localization [39,40,41]. For the lasing spectra (Figure 5b,c), we observe a similar pattern down to approximately 160 K, at which point the laser switches off. Spectra measured for lower temperatures are significantly red-shifted, which indicates a decrease in the screening of the quantum-confined Stark effect. Apparently, we face a sudden decrease in carrier concentration in the quantum wells, most likely due to the freeze-out of holes in the electron blocking layer and an increase in carrier escape from the quantum wells. 

Figure 6 presents a temperature-dependent CW operation of the same laser diode. As a similar pulse mode measurement, the slope efficiencies are calculated by using an almost all current-light (L-I) value above the threshold current. There is less non-linearity when it is compared with pulse mode results. The device lases across the entire registered temperature range, i.e., down to 77 K, albeit at the expense of increasing the threshold current and operating voltage. The threshold current increases from 46 to 264 mA, and the operating voltage at 100 mA increases from roughly 5.7 V up to 9.1 V. Additionally, after 160 K, we observed a negative differential resistivity, which is visible as a voltage drop in Figure 6a. Such behavior is usually associated with the existence of potential barriers for carrier injection. We tend to associate this barrier with potential spikes in the valence band at the position of the EBL. This barrier may increase, while the quasi-Fermi level in the EBL rises with decreasing temperature.

Interestingly, after an initial decrease with decreasing temperature, the slope efficiency sharply increases all the way to the lowest operating temperature. The temperature corresponding to the onset of the increase in threshold current and slope efficiency (160–180 K) coincides with the boundary at which the device stops to lase in pulse current measurements. The rapid increase in threshold current below 160 K is not surprising, as this is when the freeze-out of holes in the AlGaN-EBL leads to a decrease in injection efficiency [38]. 

To explain this behavior, we propose the following hypothesis: in the CW-operated laser diode, the main source of heat is the metal contact and the GaN subcontact layer due to Joule heating. In such a case, a significant gradient of temperature is present between the subcontact (Mg doped) layer and the bottom cladding layer, as Maciej Kuc demonstrated in his doctoral dissertation [42]. In Szymon Stanczyk’s doctoral dissertation [43], the gradient is demonstrated with a color map and explained in English. According to the calculations, the n-side was relatively cooler than the p-side, and the difference may reach even 25 K (at 295 K ambient temperature). The difference might be increased while lowering the ambient temperature. The local increase in temperature in top layers helps to keep the Mg atoms close to the interface ionized. The presence of this hole reservoir allows for sustaining laser diode operation at low temperatures despite the increasingly poor performance of the EBL (in the cooler zone). The increase in slope efficiency could be related to faster thermalization and an easier capture of electrons injected from the relatively colder n-type layers into the quantum wells. The increase in slope efficiency at lower temperatures may be also related to a reduction in optical losses. Szymon Stanczyk [43] demonstrated that there is a clear reduction in optical losses with temperatures in classical Mg-doped nitride lasers; however, this measurement was not extended to cryogenic temperatures, and the physical mechanism behind this behavior is not clear. In contrast with our results, Sizov et al. [44] demonstrated the optical loss of the Mg-doped GaN that is independent from temperature and the ionization of Mg dopant. In the same paper, a significant change in losses is measured between activated Mg and non-activated Mg-H complexes. However, there is no reason for the Mg dopant to become passivated during the cooling of our lasers. In our future studies, we plan to perform additional studies of the effect of low temperature on optical losses. The optical losses can be estimated by Hakki–Paoli gain measurement. However, currently, due to technical limitations of the measurement setup, we were not able to perform such experiments.

An alternative approach to overcome the EBL problem at cryogenic temperature could be the use of an inverted structure (n-tunnel junction-p-n) proposed by Chlipala et al. [23]. Because of polarization direction, the quantum well and quantum barrier interface will bend up and the electron cannot overshoot the well. However, in such kind of structure, the tunnel junction needs to be grown below the core of the device structure. In such a case, the activation of the Mg-doped layer of the tunnel junction becomes a problem. The inverted structure is made by Chlipala et al. [37], which is grown by MBE with no need for a Mg activation process. For MOCVD growth, the activation process is required, which leads to extra processing steps such as etching down to tunnel junction to allow for out-diffusing H during an annealing process.

In Figure 7, we present the spectral characteristics of the emission from our polarization-doped laser. The spectra are measured at driving currents below the threshold (Figure 7a) and above the threshold (Figure 7b). Figure 7a,c show that the emission wavelength weakly depends on temperature, similar to the spectra observed in the pulse current experiment (S-shape behavior). Note that the spectra shown in Figure 7b are recorded at various drive currents. This is because the threshold current increases with temperature, and to keep the device above the threshold, we need to increase the current. Generally, we aim to operate our laser at a constant current distance above the threshold. Because of huge changes in the threshold current, we make the measurements at around 50 mW of optical power of the laser diode for each temperature.

We observed that the behavior of the wavelengths of lasing and spontaneous emission is similar. In the case of lasing, we saw a small red shift of emission at temperatures below 140 K, which can be associated with higher drive currents, resulting in higher junction temperatures. However, this effect is relatively small. The boundary between the two different regions appears at 150 K, which is similar to the jump observed in Figure 5c for laser emission. 

Finally, to measure the level of self-heating, we performed thermal resistance measurements using the technique described in the paper of Scheibenzuber et al. [45]. The method is based on the dependence of a refractive index (and consequently cavity mode position) on temperature. The thermal resistance of our sample is estimated to be 50 K/W, as shown in Figure 8. Assuming that the thermal resistance does not change with temperature, the minimum internal temperature of the laser at threshold currents is around 200 K, which is higher than the hole freeze-out temperature. This explains the possibility of laser diode operation at very low temperatures in CW mode, in contrast with the suspension of lasing in the pulsed operation.

## 4. Conclusions

In this work, we demonstrate the successful operation of p-type polarization-doped light emitting diodes and laser diodes at cryogenic temperatures.

The polarization-doped LEDs can operate even down to 20 K. However, due to a design without an AlGaN electron blocking layer, the optical emission efficiency at low temperatures seems to be limited by electron overshooting through the quantum wells. Our studies suggest that the GaN:Mg subcontact layer keeps satisfactory work parameters even at the lowest operating temperature (20 K).

In the case of laser diodes, they maintain very good parameters over a broad temperature range (as for InGaN lasers): between 295 K and 200 K, with only small variations in threshold current and slope efficiency. For lower temperatures, the limiting factor is the freeze-out of the magnesium-doped AlGaN electron blocking layer, becoming a strong effect at 160 K and below. This is probably related to the Joule heating in the area of the metal–semiconductor interface. However, still, we were able to demonstrate lasing of the polarization-doped laser diode down to 77 K ambient temperature. Unexpectedly, the slope efficiency of the laser diode increases below 160 K, which we attribute to the improved electron capture in the quantum wells. To further improve the low-temperature operation of polarization-doped InGaN laser diodes, we should find a way to eliminate the magnesium-doped EBL and replace it with a uniquely polarization-doped layer. 

## Figures and Tables

**Figure 1 materials-17-04502-f001:**
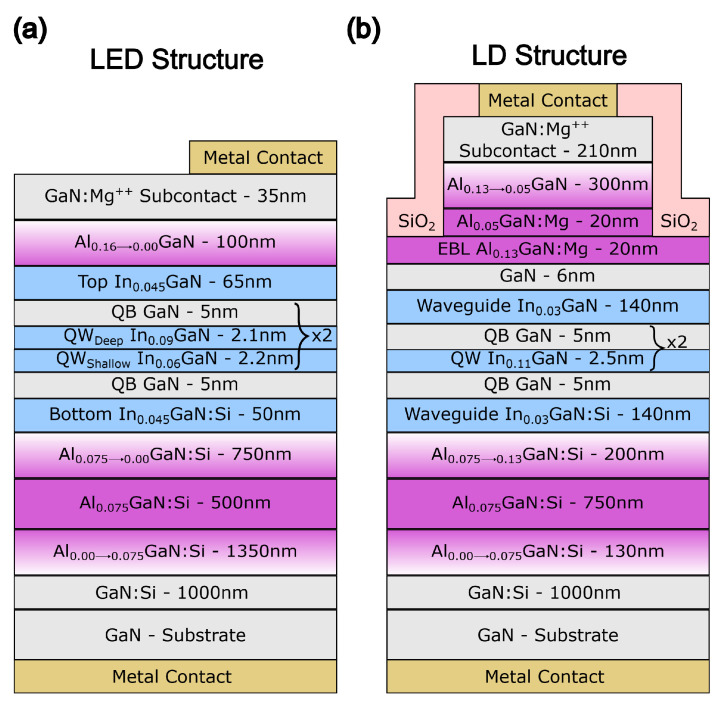
Structure of LED (**a**) and laser (**b**) with polarization-doped p-cladding layer.

**Figure 2 materials-17-04502-f002:**
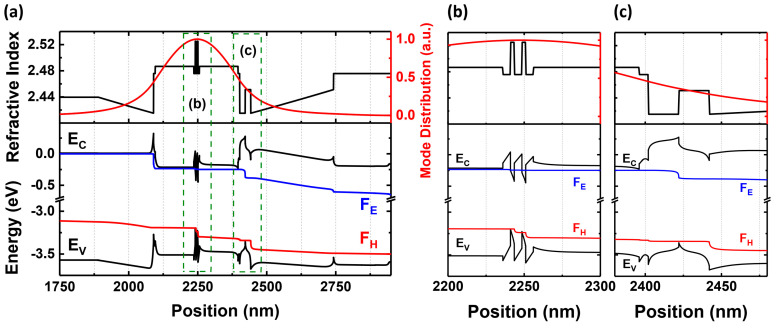
Energy − band diagram (@ 3.5 V) and refractive index of the laser structure (**a**) zoomed at an active area (**b**) and EBL area (**c**). The regions corresponding to (**b**,**c**) are marked by green dashed rectangles in the picture (**a**). Vertical axes are identical for all graphs.

**Figure 3 materials-17-04502-f003:**
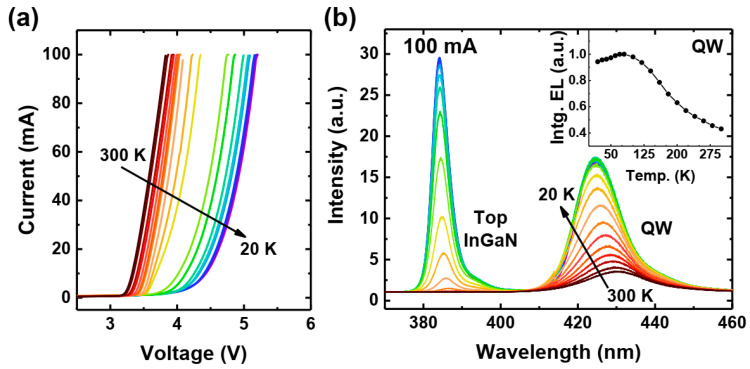
Polarization doped LED’s I–V measurement (**a**); EL measurement at 100 mA (**b**).

**Figure 4 materials-17-04502-f004:**
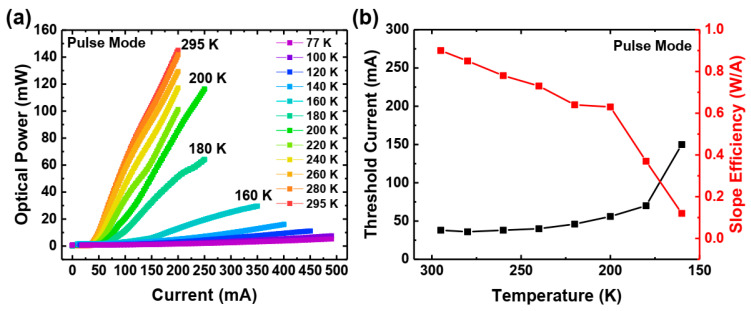
Polarization-doped laser structure’s L–I–V measurement in pulse mode (**a**). Threshold currents and slope efficiencies in pulse mode (**b**).

**Figure 5 materials-17-04502-f005:**
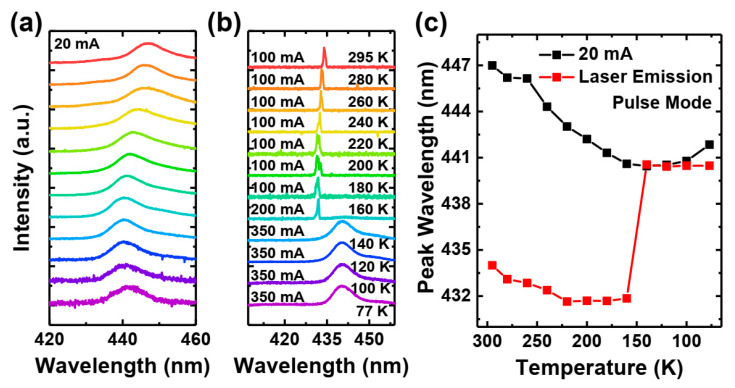
Temperature dependence pulse mode EL spectra of polarization-doped laser structure at low current (**a**), high currents (**b**), and their peak wavelength (**c**).

**Figure 6 materials-17-04502-f006:**
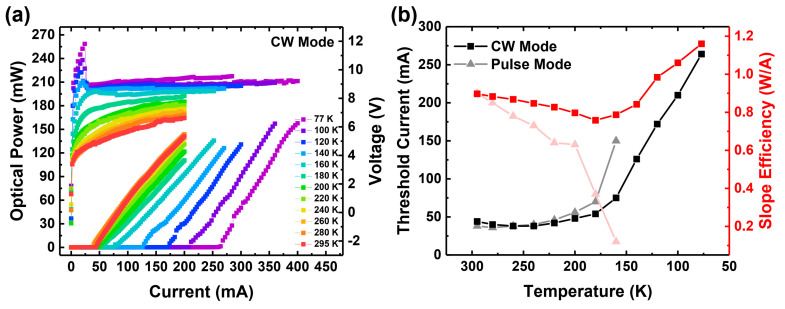
L–I–V measurement in CW mode of the polarization-doped laser structure (**a**). Threshold currents and slope efficiencies in CW mode (**b**), with pulse mode results for reference.

**Figure 7 materials-17-04502-f007:**
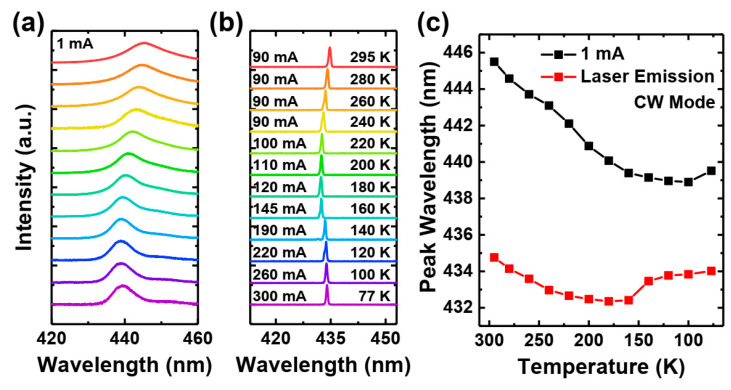
Temperature dependence EL spectra of polarization-doped laser structure at 1 mA (**a**), above the threshold current (**b**), and their peak wavelength (**c**) in CW mode.

**Figure 8 materials-17-04502-f008:**
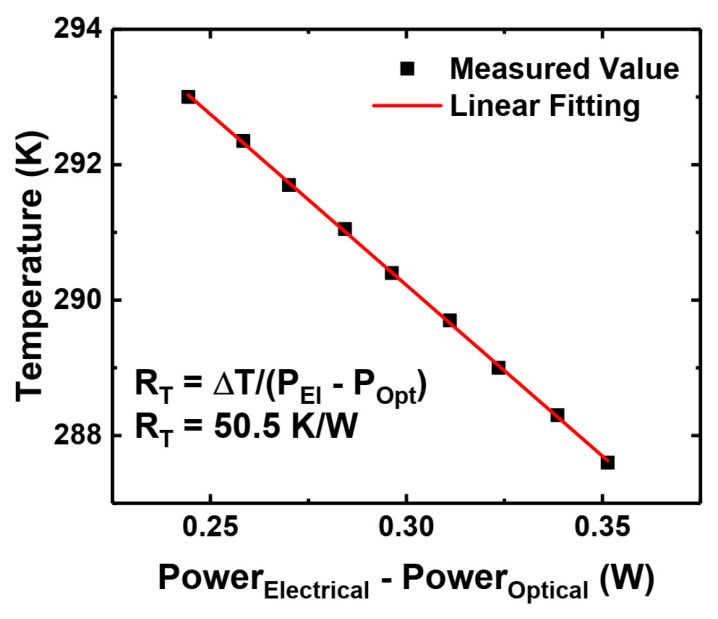
Thermal resistance of polarization doping laser structure.

## Data Availability

Data are available upon contacting the corresponding author.

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
