# Peer review of "Polarization-Doped InGaN LEDs and Laser Diodes for Broad Temperature Range Operation"

_materials, 2024, doi:10.3390/ma17184502_

Round 1

Reviewer 1 Report

Comments and Suggestions for Authors

This paper reports an interesting study on InGaN LEDs and laser diodes with polarization-doped hole injection layers. The main conclusion was that the devices could be operated at cryogenic temperatures by using the polarization-doped hole injection structure. I feel that the paper is publishable in MDPI Materials but a couple of revisions are required.

1) The active layer of the LED is very different from the conventional design of an InGaN/GaN blue LED. More description should be given to the design of the LED structure. Why a “two-layer” (QWshallow + QWdeep) design for the well layer is used?

2) Page 4, line 165-166. “Above 200 K, we started to observe a secondary emission peak with energy corresponding to the top InGaN layer.” However, from Fig. 2(b), strong emission from the top InGaN layer was observed at low temperatures (even at 20 K). “Above 200 K” seems to be the mistake of “Below 200 K”. Please confirm.

3) The inset of Fig. 2(b). Does the “integrated EL intensity” represent only the integrated intensity of the QW peak? It is clear from the emission spectra that a considerable portion of the injected carriers are trapped by the top In0.045GaN layer. This would influence the temperature dependence of the integrated EL intensity of the QW peak. Please clarify.

4) Measurement of the temperature dependence of the laser diode. The authors mentioned in page 5 that “the cryostat’s cooling capacity can approach the energy dissipation within the device”. Have the authors compared the cooling capacity of the cryostat with the injected electrical power? Does this mean that the actual temperature of the devices under CW operation could be much higher than the measured (or setting) temperatures? Some comments on the influence of the cryostat’s cooling capacity on the lasing characteristics under CW operation are desired. 

Author Response

We attached a marked revised version of the manuscript. Changes related to Reviewer 1 comments are marked by cyan highlight. Changes related to Reviewer 2 comments are marked by yellow highlight. Technical changes are marked by gray highlight.

Comments 1: The active layer of the LED is very different from the conventional design of an InGaN/GaN blue LED. More description should be given to the design of the LED structure. Why a “two-layer” (QWshallow + QWdeep ) design for the well layer is used?

Response 1: Thank you for your kind comment. This type of quantum well design (staggered QW) was introduced by Zhao et al. [1] and Arif et al. [2]. It is introduced to increase the overlap of electron and hole wavefunctions which are separated in InGaN/GaN QWs due to the built-in electric field (Quantum Confined Stark Effect), and by this increase the recombination rate. But, this design is mostly beneficial for QWs with higher In content (green and longer wavelengths). Which is why after performing a number of tests (that included the LED sample) we dropped the use of this design for QWs with In content below 17% including the presented here laser diode structure. We added related comment in Page 4 between 152 to 156 lines.

[1]        H. P. Zhao et al., “Design and characteristics of staggered InGaN quantum-well light-emitting diodes in the green spectral regime,” IET Optoelectronics, vol. 3, no. 6, pp. 283–295, Dec. 2009, doi: 10.1049/iet-opt.2009.0050.

[2]        R. A. Arif, Y. K. Ee, and N. Tansu, “Polarization engineering via staggered InGaN quantum wells for radiative efficiency enhancement of light emitting diodes,” Appl Phys Lett, vol. 91, no. 9, 2007, doi: 10.1063/1.2775334.

Comments 2: Page 4, line 165-166. “Above 200 K, we started to observe a secondary emission peak with energy corresponding to the top InGaN layer.” However, from Fig. 2(b), strong emission from the top InGaN layer was observed at low temperatures (even at 20 K). “Above 200 K” seems to be the mistake of “Below 200 K”. Please confirm.

Response 2: Thank you for your kind consideration. You are right - we updated the part to “below 200 K”. We made the change about  this comment in Page 5 at 192 line.

Comments 3: The inset of Fig. 2(b). Does the “integrated EL intensity” represent only the integrated intensity of the QW peak? It is clear from the emission spectra that a considerable portion of the injected carriers are trapped by the top In GaN layer. This would influence the temperature dependence of the integrated EL intensity of the QW peak. Please clarify.

Response 3: We are grateful for this suggestion. The integrated EL intensity only contains the QW emissions part, and we provide additional explanations for this. We added related comment  in Page 5 between 187  to 189 lines.

Comments 4: Measurement of the temperature dependence of the laser diode. The authors mentioned in page 5 that “the cryostat’s cooling capacity can approach the energy dissipation within the device”. Have the authors compared the cooling capacity of the cryostat with the injected electrical power? Does this mean that the actual temperature of the devices under CW operation could be much higher than the measured (or setting) temperatures? Some comments on the influence of the cryostat’s cooling capacity on the lasing characteristics under CW operation are desired

Response 4: Thank you very much for your comment. The temperature is measured by a thermocouple connected to the cryostat’s cold finger. The laser diode is directly attached to the cryostat's cold finger, and consequently, the laser package should have approximately the same temperature. If the cooling power of the cryostat were lower than the heat dissipated, we would not have been able to stabilize the cryostat temperature, which was not the case. However, during continuous-wave (CW) operation, we can expect a temperature gradient starting from the coldest part (the package), through the submount, laser substrate, and then the active area and p-type cladding, ending at the p-side electrode. The hottest part of the laser will be its p-contact and p-cladding, both located near the surface of the structure. Many of the “strange” physical effects arise from the presence of these temperature gradients. In the case of pulse operation, as long as the pulses are shorter than approximately 0.5 μs, self-heating becomes negligible, and we can assume the entire structure to have a uniform temperature. To understand the temperature change caused by electrical power, we present the thermal resistance in Figure 7. The difference of temperature between ambient and inside of the chip are difficult to directly estimate and set a clear drawback for our lasers from the point of view of future material studies at cryogenic temperatures. But from the application point of view it is not a problem, as the important factor is the value of the ambient temperature for which the device can operate, not the temperature of the chip. We added related comment in Page 6 between 203  to 208 lines and between 213  to 215 lines

Reviewer 2 Report

Comments and Suggestions for Authors

The paper is devoted to the study of III-N light-emitting structures created using the polarization doping method. The work leaves a good impression of a systematically conducted study. However, there are a number of questions and comments.

1.    In motivating their work, the authors point out that p doping does not work well at cryogenic temperatures due to the high activation energy of Mg. This motivation model lacks a logical connection - why is it necessary to obtain light emitters operating at low temperatures? Is this due to the peculiarities of their highly specialized applications and operating conditions, or is it because radiative recombination in the active region is more efficient? An explanation is needed.
2.    The key to this work is the band diagram of heterostructures. Alas, the authors do not provide a single such diagram. Without them, it is extremely difficult to understand the structure of the proposed structures. It is recommended to provide such diagrams with the necessary explanations. (The authors refer to their previous work [25], but unfortunately, a complete band diagram that would take into account all polarization effects is not presented there either).
3.    For a better understanding of the working structure, it is recommended to provide a drawing with the etching scheme and contact arrangement.
4.    What is the reason for choosing the temperature ranges for measuring LED and LD structures (20-300 and 77-295)? What prevented measuring LD up to 20K?
5.    The results of measuring the optical power of LD in pulse mode at different temperatures are somewhat counterintuitive - as the temperature decreases, a decrease in intensity and an increase in the threshold current are observed. Unfortunately, I did not find a clear explanation for this phenomenon in the text.

Author Response

We attached a marked revised version of the manuscript. Changes related to Reviewer 1 comments are marked by cyan highlight. Changes related to Reviewer 2 comments are marked by yellow highlight. Technical changes are marked by gray highlight.

Comment 1: In motivating their work, the authors point out that p doping does not work well at cryogenic temperatures due to the high activation energy of Mg. This motivation model lacks a logical connection - why is it necessary to obtain light emitters operating at low temperatures? Is this due to the peculiarities of their highly specialized applications and operating conditions, or is it because radiative recombination in the active region is more efficient? An explanation is needed.

Response 1: We thank the Reviewer for noticing this lack of explanation. We have three main motivations behind the presented studies. 1) From a purely practical point of view, we direct our attention to the applications in rough environments where the operation temperature may significantly vary even towards very low temperatures. A good example is space-type visible light communication systems for satellites. 2) Laser diodes operating at cryogenic temperature will be useful for various cryogenic studies, as the excitation light source may be placed directly inside the cryostat. 3) According to our knowledge, the polarization-doped laser diodes seem to be the only solution to study the temperature-dependent material parameters such as material gain down to cryogenic temperatures. So, our goal is also expanding the understanding of the material properties of nitride devices. We added related comment in Page 2 between 47 to 55 lines.

Comment 2:  The key to this work is the band diagram of heterostructures. Alas, the authors do not provide a single such diagram. Without them, it is extremely difficult to understand the structure of the proposed structures. It is recommended to provide such diagrams with the necessary explanations. (The authors refer to their previous work [25], but unfortunately, a complete band diagram that would take into account all polarization effects is not presented there either).

Response 2: We are grateful for this suggestion. We added the energy-band structure and refractive index of laser structure and necessary explanations. We added in Figure 2 in page 4 and this comment in Page 3 between 135  to 142 lines.

Comment 3:  For a better understanding of the working structure, it is recommended to provide a drawing with the etching scheme and contact arrangement.

Response 3: Thank you for your kind suggestions. We updated Figure 1 in page 4 to demonstrate contact arrangements. Please note that the extra 6 nm GaN between waveguide and EBL layer in laser is technical layer. We added as it can be noticed in added new Figure 2.

Comment 4:  What is the reason for choosing the temperature ranges for measuring LED and LD structures (20-300 and 77-295)? What prevented measuring LD up to 20K?

Response 4: Thank you very much for your kind comment. The reason for this difference lies in the technical limitations. As mentioned in the manuscript, the LED device was mounted on a copper plate and the laser device was mounted on the TO-56 package. The measurement equipment for cryogenic temperatures is different, and the equipment at 20 K is not suitable for the TO-56 package. The laser device can be measured on a copper plate; however, it is not comparable to the standard applications. We added this comment in Page 6 between 201 to 203 lines.

Comment 5:  The results of measuring the optical power of LD in pulse mode at different temperatures are somewhat counterintuitive - as the temperature decreases, a decrease in intensity and an increase in the threshold current are observed. Unfortunately, I did not find a clear explanation for this phenomenon in the text.

Response 5: Thank you very much for your valuable comment. The change in the threshold currents and slope efficiency should be caused by two effects. Firstly, the ionization of Mg dopant at the EBL. The hole concentration at the electron blocking layer (EBL) is directly proportional to the efficiency of the blocking layer. With the reduction of the temperature, the concentration of holes is reduced. Secondly, a typical problem of nitride structures is the low mobility of holes, which may be blocked by a potential barrier at the electron blocking layer. As the temperature is reduced, overcoming this barrier becomes a bigger problem and the efficiency of hole injection to the QWs is reduced. This mechanism is independent of the usage of polarization doping. We added related comment in Page 6 between 216  to 223 lines.

Round 2

Reviewer 1 Report

Comments and Suggestions for Authors

The authors have properly adressed all my previous comments.  I don't have further comments. I think the paper is now ready for publication.  

Reviewer 2 Report

Comments and Suggestions for Authors

Authors takes good response for all coments. The paper can be published.

The new fig. 2 with the band diagram is not very clear for understanding. I recomend to separate band diagram and refractive index trends.